# Multi-Target and Multi-Phase Adjunctive Cerebral Protection for Acute Ischemic Stroke in the Reperfusion Era

**DOI:** 10.3390/biom14091181

**Published:** 2024-09-20

**Authors:** Min Zhao, Jing Wang, Guiyou Liu, Sijie Li, Yuchuan Ding, Xunming Ji, Wenbo Zhao

**Affiliations:** 1Department of Neurology, Xuanwu Hospital, Capital Medical University, Beijing 100053, China; 2Beijing Institute for Brain Disorders, Capital Medical University, Beijing 100069, China; 3Department of Emergency, Xuanwu Hospital, Capital Medical University, Beijing 100053, China; 4Beijing Key Laboratory of Hypoxic Conditioning Translational Medicine, Xuanwu Hospital, Capital Medical University, Beijing 100053, China; 5Department of Neurosurgery, Wayne State University School of Medicine, Detroit, MI 48201, USA; 6Department of Neurosurgery, Xuanwu Hospital, Capital Medical University, Beijing 100053, China

**Keywords:** acute ischemic stroke, cerebral protection, ischemic penumbra, reperfusion injury

## Abstract

Stroke remains the leading cause of death and disability in some countries, predominantly attributed to acute ischemic stroke (AIS). While intravenous thrombolysis and endovascular thrombectomy are widely acknowledged as effective treatments for AIS, boasting a high recanalization rate, there is a significant discrepancy between the success of revascularization and the mediocre clinical outcomes observed among patients with AIS. It is now increasingly understood that the implementation of effective cerebral protection strategies, serving as adjunctive treatments to reperfusion, can potentially improve the outcomes of AIS patients following recanalization therapy. Herein, we reviewed several promising cerebral protective methods that have the potential to slow down infarct growth and protect ischemic penumbra. We dissect the underlying reasons for the mismatch between high recanalization rates and moderate prognosis and introduce a novel concept of “multi-target and multi-phase adjunctive cerebral protection” to guide our search for neuroprotective agents that can be administered alongside recanalization therapy.

## 1. Introduction

Acute ischemic stroke (AIS) results from cerebral artery occlusion, and, therefore, revascularization therapy is regarded as the most effective therapy [1]. For 30 years ago, intravenous thrombolysis has been used for reperfusion therapy of AIS [2,3,4]. Furthermore, in the past decade, endovascular thrombectomy has been demonstrated to be effective in recanalizing large cerebral artery occlusion [5]. Many studies have demonstrated that for AIS, due to large vessel occlusion, endovascular thrombectomy could achieve a favorable recanalization rate of up to 88% or much higher, greatly improving the patients’ functional outcomes [6,7,8,9]. However, despite the high revascularization rates achieved in patients with AIS, there exists a significant discrepancy between these rates and clinical outcomes, with approximately 50~60% of patients experiencing poor functional outcomes or even death within 90 days after reperfusion treatments [7,10,11,12,13].

Currently, research in the field of revascularization therapy for AIS has focused on how to further improve functional outcomes [14,15]. Cerebral protection has been considered the most promising strategy, and thousands of neuroprotective methods have been developed or discovered, hundreds of them being translated into clinical practice [16,17]. However, the clinical efficacy of cerebral protective treatment has rarely been shown to provide clinical benefits [18,19,20,21,22].

This review summarized several promising neuroprotective methods that have the potential to improve outcomes for patients with AIS. It thoroughly analyzed the reasons underlying the discrepancy between the high recanalization rates achieved and the mediocre prognosis observed, highlighting the need for a more comprehensive approach. To bridge this gap, this review introduced the novel concept of “multi-target and multi-phase adjunctive neuroprotection”, which guides our search for neuroprotection that can be effectively used alongside recanalization therapy.

## 2. Reasons for Mediocre Prognosis

Neuroprotective strategies primarily aim to arrest the ischemic cascade, mitigate reperfusion injury, activate endogenous collaterals, and foster the development of immunomodulators [23]. Over the past several decades, trials of cytoprotective therapy have focused on single-target pathophysiological mechanisms. However, these have shown limited efficacy in improving functional outcomes in clinical trials [20,21,24]. Nerinetide (NA-1) was once regarded as a promising neuroprotective drug that could improve clinical outcomes for patients with AIS [23]. This optimism stemmed from compelling evidence discovered in studies conducted over the past decade in rodents [25,26] and primate animals [27], indicating that NA-1 could mitigate the ischemic core and improve functional outcomes. However, the ESCAPE-NA1 (https://clinicaltrials.gov/study/NCT02930018, accessed on 18 September 2024) study revealed that while the administration of NA-1 before mechanical thrombectomy (MT) for patients with AIS is safe and feasible, it does not enhance clinical outcomes at 90 days [20]. Following a subgroup analysis suggesting that the efficacy of NA-1 might be influenced by recombinant tissue plasminogen activator (rt-PA), ESCAPE-NEXT (https://clinicaltrials.gov/study/NCT04462536, accessed on 18 September 2024) conducted an additional investigation. However, the results presented at the 15th World Stroke Congress regrettably indicated that the combination of NA-1 with endovascular thrombectomy (EVT) did not enhance clinical outcomes compared to EVT alone. Notably, none of the patients in this study received rt-PA.

The pathophysiology of ischemic core expansion and reperfusion injury encompasses intricate, multi-faceted mechanisms, principally oxidative stress, excitotoxicity, calcium dysregulation, and inflammation. The ischemic penumbra, comprising salvageable brain tissue, gradually deteriorates into irreversibly damaged tissue before reperfusion treatments, leading to the enlargement of the ischemic core [28]. Collateral flow serves as a mitigating factor, partially alleviating distal arterial embolism and hypoperfusion that arise from severe proximal arterial thrombosis. Specifically, the collateral blood supply to the penumbra region plays a pivotal role in determining the progression rate of the ischemic process. After reperfusion, restoring blood flow to ischemic brain tissue curtails the expansion of the ischemic core but concurrently triggers secondary injury [29]. The secondary reperfusion injuries encompass the generation of reactive oxygen species and the activation of inflammatory and immune responses, which can culminate in cell death, thereby augmenting the infarct volume. Furthermore, they disrupt the blood–brain barrier, precipitating cerebral edema and intracranial hemorrhage [30,31,32]. Moreover, the no-reflow phenomenon has been identified as another significant factor contributing to poor functional recovery in stroke patients [33]. The primary reason for the failed trials may be associated with the absence of comprehensive and effective neuroprotection treatment strategies that target the multi-faceted pathophysiological processes of AIS and the various phases of reperfusion therapy while integrating with revascularization therapy [23,34,35]. Therefore, multi-target and multi-phase adjuvant neuroprotection may be an attractive prospect to achieve comprehensive and effective neuroprotection.

## 3. Non-Pharmacological Neuroprotective Methods

Non-pharmacological cerebral protective approaches generally play their role in stabilizing infarct volume and protecting the ischemic penumbra through multi-targets on the processes of pathological and physiological changes in AIS (Table 1). In addition, physical neuroprotective strategies function without relying on blood flow as a carrier, in contrast to drugs administered to the ischemic site through blood circulation.

### 3.1. Remote Ischemic Conditioning

Remote ischemic conditioning (RIC) is an easy-to-use, cheap, and well-tolerated non-invasive therapy for AIS. It entails the application of several cycles of ischemia and reperfusion to a tissue or organ, such as the use of a blood pressure cuff on a limb [36]. The exact mechanism underlying the protective effects of RIC on the brain remains unclear, but it has been demonstrated to effectively inhibit the ischemic cascade and reperfusion injury and concurrently enhance cerebral flow through various pathways that encompass humoral, neural, and immune mechanisms [37,38]. This comprehensive approach ensures the limitation of ischemic core growth and protection of the ischemic penumbra. The protective effects of RIC against reperfusion injury are multi-faceted, mediated by an intricate network of molecular signaling pathways that converge upon pivotal transcription factors regulating cell survival and apoptosis. Notably, the reperfusion injury salvage kinase and the survivor activating factor enhancement pathways are well-defined mechanisms [39]. Evidence indicates that RIC possesses the capability to not only inhibit the activation of microglia and astrocytes after an acute ischemic stroke but also hinder the recruitment of circulating peripheral immune cells into the ischemic brain, thereby mitigating the inflammatory response and promoting neuroprotection [40]. RIC can potentially improve collateral flow by preventing the narrowing of collateral vessels and facilitating the formation of vascular branches [41]. Chao et al. have demonstrated the neurogenic pathway through transecting the nerve, which had undergone remote ischemic preconditioning. Their finding revealed that this manipulation diminishes the cerebral protective benefits typically conferred by remote ischemic conditioning [42]. A variety of humoral factors, such as hydrophobic peptides, opioid peptides, adenosine, and others, have been shown to exhibit a relationship with RIC [43].

Research on RIC has shown promising results in recent years. In focal cerebral ischemia models, the application of RIC before cerebral ischemia, during ischemia, or/and after reperfusion therapy has significantly reduced the final infarct volume without causing related adverse events [44]. Early clinical trials, typically involving less than 100 patients, showed the safety of RIC as an adjunctive to reperfusion in AIS [45,46,47]. Recently, several large-scale randomized clinical trials have been successfully conducted, focusing on exploring the efficacy of RIC in AIS, and the results have been encouraging, significantly enhancing the likelihood of achieving excellent neurological function [47,48]. Hougaard et al. conducted an open-label, blinded outcome, proof-of-concept study to investigate the effect of prehospital RIC as an adjunctive treatment to intravenous therapy in patients with suspect AIS [49]. During transportation, four cycles of RIC stimulus were administered by ambulance staff, and if the procedure was not fully completed, it was terminated upon arrival at the stroke unit. Though the primary endpoints referring to the volume of the perfusion–diffusion mismatch after 1 month were not significantly different among the various groups, subgroup analysis indicated that the RIC group had lower admission on the National Institute of Health Stroke Scale (NIHSS) scores than controls. RICAMIS (https://clinicaltrials.gov/study/NCT03740971, accessed on 18 September 2024), a multicenter clinical trial involving 1893 patients with moderate AIS, tested the efficiency of RIC as an adjunct to usual care compared with usual care alone. The main outcome, excellent neurologic functional outcome at 90 days, was significantly different among groups, which probably indicated that RIC increased the functional outcomes in AIS patients [48]. However, RESIST (https://clinicaltrials.gov/study/NCT03481777, accessed on 18 September 2024), including 1500 patients with symptom onset for less than 4 h, aims to assess the effect of RIC before and during hospital in patients with acute stroke. The results of the primary outcome, an excellent functional outcome at 90 days, are neutral [50].

In accordance with the aforementioned studies, RIC emerges as a promising area of AIS research, and numerous clinical trials focus on RIC currently underway. SERIC-EVT (https://clinicaltrials.gov/study/NCT04977869, accessed on 18 September 2024) trial and SERIC-IVT (https://clinicaltrials.gov/study/NCT04980625, accessed on 18 September 2024) trial intend to evaluate the efficacy of RIC for AIS patients after reperfusion therapy [51,52]. Another multicenter randomized control trial (https://center6.umin.ac.jp/cgi-open-bin/ctr/ctr_view.cgi?recptno=R000052753, accessed on 18 September 2024) focuses on the efficacy of RIC among different NIHSS scores. Furthermore, TRICS-9 (https://clinicaltrials.gov/study/NCT04400981, accessed on 18 September 2024) has been registered to assess RIC in patients with AIS within 9 h of symptom onset, who were not eligible for recanalization therapies.

### 3.2. Normobaric Hyperoxia

Normobaric hyperoxia (NBO) therapy refers to the administration of oxygen with an increased concentration or flow rate to patients under standard atmospheric pressure, utilizing oxygen hoods or masks as the delivery system. Although the precise cerebral protective mechanisms of NBO remain elusive, the enhancement of neuronal metabolism, augmentation of oxygen content in brain tissues, reduction in blood–brain barrier (BBB) damage, and enhancement of cerebral blood flow are likely significant factors associated with its neuroprotective effects [53,54,55].

In animal studies, NBO has been shown to improve oxygen supply to ischemic tissue, temporarily stabilize the ischemic penumbra, and extend the therapeutic window for revascularization [53,56]. The results of NBO in early clinical research are controversial, likely with respect to trial design, involving delayed administration of NBO, low NBO flow or concentration rates, start or duration time of NBO, time late or lack reperfusion of patients, and small sample size [57,58,59,60].

Recent clinical trials have further validated the safety and feasibility of NBO combined with reperfusion for patients with AIS [61,62]. Li et al. aimed to examine the effect of NBO combined with EVT in AIS patients. Patients were delivered 100% oxygen before EVT, and the primary endpoint assessed the volume of infarct through MRI within 24–48 h after reperfusion. The results demonstrated that NBO, as an adjuvant to EVT, exhibited a favorable safety profile and held the potential to reduce infarct volume at the early stages of AIS, thereby improving the functional outcomes for patients. However, it is worth noting that this study was conducted at a single center and involved a relatively small sample size [62]. Similarly, Zhe et al. and Li et al. verified the safety and efficacy of NBO as a supplementary therapy of reperfusion. Specifically, NBO has been shown to reduce infarct volumes, improve functional outcomes, and decrease mortality, all without increasing the incidence of symptomatic intracranial hemorrhage (sICH) [63,64]. However, Cheng et al. conducted a prospective randomized controlled study focusing on the cerebral protection effect of NBO for patients with posterior circulation stroke, in which the results did not provide statistical evidence supporting the neuroprotective effect of NBO in terms of improving functional outcomes or reducing mortality rates [65]. Since NBO reveals greater efficacy when administered as early as possible after symptom onset, PROOF (https://clinicaltrials.gov/study/NCT03500939, accessed on 18 September 2024) trial, a phase II trial, was designed to prove the value of NBO therapy in AIS patients within 6 h of notice until recanalization. Future research should consider larger multicenter clinical trials to further validate the observed benefits of NBO [66].

### 3.3. Sphenopalatine Ganglion Stimulation

The sphenopalatine ganglion (SPG), a crucial structure for intracranial blood vessel vasodilation, regulates ipsilateral cerebral blood flow through arterial vasodilatation. The primary mechanism of sphenopalatine ganglion stimulation (SPGS) appears to involve the dilation of arteries and augmentation of ipsilateral cerebral blood flow through the modulation of parasympathetic innervation [67]. Preclinical studies on acute anterior circulation ischemic stroke have shown that stimulating the sphenopalatine ganglion can temporarily stabilize the ischemic penumbra, safeguard the blood–brain barrier, and improve the prognosis of neurological function [68,69].

Recent clinical research indicates that among patients with cortical involvement at presentation, SPG stimulation is likely to improve functional outcomes. Clinical studies tend to favor the application of trans-oral electrical stimulation to the sphenopalatine ganglion, achieved through the use of implanted devices that are precisely injected into the pterygopalatine canal in close proximity to the sphenopalatine ganglion. This procedure is considered a minimally invasive technique, often performed under local anesthesia. The ImpACT-24B trial (https://clinicaltrials.gov/study/NCT00826059, accessed on 18 September 2024) was designed to demonstrate the safety and efficacy of SPGS in patients with anterior-circulation AIS within 8 to 24 h of symptom onset. The primary efficacy endpoint is the difference between intervention and sham groups in 3-month functional outcomes. The subgroup with confirmed cortical involvement (CCI) exhibited statistically significant improvements in efficacy endpoints. Moreover, the relationship between the intensity of SPGS and its efficacy in the CCI subgroup follows an inverse U-shaped pattern, suggesting that a lower to mid-level stimulation may be the most effective approach [70]. Saver et al. conducted a single SPGS trial, including patients with anterior circulation ischemic stroke exhibiting arm weakness within 24 h of symptom onset. The efficacy of SPGS was evaluated through the measurement of volumetric blood flow in the ipsilateral common carotid artery via ultrasound and the assessment of grasp and pinch strength in the affected hand both before and after the stimulation. The results demonstrated that SPG stimulation enhanced cerebral blood flow, resulting in an improvement in the motor function of the hand [71]. Notably, none of the aforementioned several large clinical trials involved the use of reperfusion therapies. Hence, SPGS has nearly no safety concerns and holds considerable promise as a cerebral protection adjunct for available reperfusion therapies to improve clinical outcomes.

### 3.4. Selective Brain Cooling Methods

Selective brain cooling (SBC) methods aim to reduce the local or overall temperature of the brain while minimizing the impact on overall body temperature, substantially avoiding the drawbacks of systemic hypothermia [72]. Surface brain cooling and endovascular cooling are two main strategies.

Endovascular cooling relies on invasive catheters that channel chilled saline or endovascular cooling devices to reach the ischemic core more quickly and precisely compared to surface cooling. An early clinical study conducted by Choi et al. was designed to examine the safety and feasibility of endovascular brain cooling in 18 patients who were undergoing cerebral angiography. The results indicated that brain cooling was safe and feasible, for the brain temperature decreased more rapidly than systemic temperature, and there were no signs of vascular spasm or neurologic deficits [73]. Similarly, another study by Chen et al. indicated that intra-arterial selective cooling infusion (IA-CSI) was safe and feasible for patients with large-vessel occlusion [74]. After that, Peng et al. enrolled 26 patients with acute middle cerebral artery occlusion mainly to demonstrate the efficacy of arterial thrombolysis combined with 4 °C saline IA-CSI. The results suggested that intra-arterial mild hypothermia could reduce infarct volume and improve functional outcomes of patients [75]. In 2018, Wu et al. launched a larger prospective cohort study to assess the functional efficacy of IA-CSI in patients undergoing mechanical thrombectomy (MT). Results indicated that IA-CSI at pre- and post-reperfusion promoted functional independence at 90 days compared to patients who received thrombectomy therapy only, but the discrepancy was not statistically significant [76]. Currently, more clinical trials are underway to further test the effect of IA-SCI adjunct to MT in AIS patients [77,78]. Notably, some limitations of IA-CSI should be noticed, particularly the occurrence of shivering and dermal vasoconstriction, which can hinder the maintenance of target temperatures. Additionally, pneumonia and cardiovascular problems associated with hypothermia have been consistently reported.

Recently, non-invasive SBC has acted as a relatively safer and less complex alternative to the invasive intra-arterial selective cooling infusion of AIS, exhibiting significant potential prospects for the future. Selective brain cooling has been proven effective in preclinical trials in slowing infarct growth in ischemic stroke by regulating cerebral metabolism, eliminating reactive oxygen species, and maintaining the integrity of the blood–brain barrier [79]. However, clinical trials have rarely achieved the same level of success as therapeutic hypothermia in animals.

Cooling helmet, one of the most convenient invasive SBC techniques, decreases the brain temperature through the head and neck surfaces. Diprose et al. designed and developed a novel cooling helmet, called the WElkins Temperature Regulation System, 2nd Gen (TRS-2), which is capable of mildly reducing brain temperature without causing severe complications. In particular, it took a significant amount of time before systemic hypothermia occurred. Therefore, a significant temperature gradient was maintained between the core and the brain, ensuring a safe and effective cooling process. However, the cooling velocity and depth remain inadequate to fulfill the requirements of clinical applications [80]. Transnasal cooling has a great anatomic advantage as the proximity of the internal carotid artery to the cavernous sinus. Using intranasal balloon catheters and 20 °C saline solution circulated for 60 min, Covaciu et al. effectively lowered the brain temperature in 10 awake volunteers by an average of 1.7 °C, with a variation of ±0.8 °C [81]. However, the majority of current transnasal cooling methods require patients to be in a sedated state [82,83].

In the future, there is a need for further exploration of the optimal velocity, duration, and depth of selective brain cooling. Non-invasive head cooling can be regarded as a crucial element of selective brain cooling, serving as a viable adjunctive treatment alongside other therapeutic hypothermia strategies.

## 4. Pharmacological Neuroprotective Methods

Certain pharmacological drugs exhibit multi-target capabilities with significant potential to improve clinical outcomes. Fingolimod, an analog of sphingosine, has demonstrated its effectiveness in reducing thrombo-inflammation, suppressing inflammatory responses, and strengthening endothelial barrier function [84]. Preclinical studies have underscored the advantageous impact of fingolimod in models of transient middle cerebral artery occlusion and thromboembolic stroke [85,86]. Fingolimod, when administered as an adjunct to standard treatments, has been shown to be safe and practical in patients with AIS, effectively limiting the expansion of the infarct volume and positively contributing to clinical outcomes at 90 days [87,88,89]. Fasudil, a non-specific inhibitor of ROCK1 and 2, effectively enhances collateral blood supply through cerebral vasodilation. A clinical trial containing 160 patients showed that fasudil treatment resulted in greater improvements in both neurological function and clinical outcomes [90]. Notably, these trials recruited patients who had not undergone reperfusion therapies, and the effect of these pharmacological approaches in patients with definitive revascularization was unexplored. Consequently, in future efforts, the neuroprotective potential of cytoprotective drugs as an adjuvant to reperfusion should be reconsidered to enhance the probability of success.

## 5. Multi-Target and Multi-Phase Adjuvant Neuroprotection

As highlighted above, the enlargement of the ischemic core, revascularization failure, and complications of reperfusion therapy significantly affect clinical outcomes for patients with AIS. Drawing from pathophysiological mechanisms and considering the different stages of reperfusion therapy, we introduce the concept of “multi-target and multi-phase adjuvant neuroprotection.” (Figure 1) The phases of neuroprotection can be predominantly grouped into three well-defined stages: preceding; concurrent with; and subsequent to recanalization therapy. Distinct objectives and considerations characterize each of these stages. In the stage preceding reperfusion, the primary focus revolves around enhancing collateral circulation and preserving the ischemic penumbra, with the aim of mitigating neuronal damage and optimizing the potential for successful recanalization therapy. During the reperfusion therapy, strategies should be used that promote recanalization of the occluded vessel and maximize the reperfusion of ischemic tissue, thereby enhancing the chances of successful functional outcomes for AIS patients. Following recanalization, the primary focus ought to be on minimizing reperfusion injury, which is of paramount importance to prevent further damage. Given the complexity of the AIS injury mechanism to brain tissue, which encompasses numerous pathophysiological processes, identifying a universal target that effectively addresses multiple processes proves challenging. Consequently, a comprehensive neuroprotective strategy for AIS patients necessitates simultaneously intervening in primary targets at every stage. Notably, when combining several neuroprotective agents, it is important to consider the potential interactions between the methods and avoid a simplistic accumulation of agents without due consideration. Multi-target approaches may offer a selectable pathway for neuroprotective therapies, with the ultimate aim of minimizing ischemic injuries. Drugs that integrate multiple distinct targets of ischemic procedures should be investigated, as this strategy has the potential to mitigate the limitations of single-target drugs, whose effects often be counteracted by the body’s compensatory mechanisms while simultaneously averting potential harmful interactions among various therapeutic interventions. Moreover, we emphasize the significance of non-pharmacological neuroprotective agents, as they inherently encompass a specific class of targets, with a heightened likelihood of their therapeutic impacts manifested in clinical outcomes.

## 6. Conclusions

The mismatch between high recanalization rates and moderate prognosis in patients with AIS is mainly attributed to ischemic core enlargement, revascularization failure, and complications related to reperfusion therapy, all of which significantly affect clinical outcomes. The concept of “multi-target and multi-phase adjunctive neuroprotection” presents a potential avenue to assist in reducing this discrepancy in the current era of highly successful reperfusion.

We propose some directions for future research. Neuroprotection treatment can be extended to include time during and even after recanalization. Administering neuroprotective drugs before recanalization, in combination with selective intra-arterial delivery during revascularization, ensures that adequate medication reaches salvageable brain tissue. Additionally, we propose the use of selective brain cooling methods, such as cooling helmets, in conjunction with intra-arterial local hypothermia to protect the ischemic penumbra. RIC can be implemented during multiple phases of reperfusion treatment. Furthermore, neuroprotective drugs should align with a strategy that combines inhibition of the ischemic cascade, mitigation of reperfusion injury, and modulation of aberrant immuno-inflammatory responses. Previous trials using neuroprotective agents failed to demonstrate a significant benefit in clinical outcomes, largely due to the lack of combination with reperfusion therapy. In the current era of reperfusion therapy, it is crucial to integrate neuroprotectants that have shown efficacy in previous preclinical studies with recanalization therapy to investigate the potential of neuroprotection as an adjunctive treatment to improve clinical outcomes.

**Table 1 biomolecules-14-01181-t001:** Clinical studies of non-pharmacological neuroprotective methods in acute ischemic stroke.

Study	N	Type of Patients	Treatment	Main Results
RIC				
RICAMIS (NCT03740971)	1893	Patients with acute moderate ischemic stroke	5 × 5 min inflations and deflations of cuff on an upper limb Cuff pressure: 200 mmHg or 25 mmHg above systolic pressure Times: within 48 h of symptom onset, and twice daily for 10 to 14 days	RIC is safe and increases the possibility of excellent function outcomes at 90 days
RISIST (NCT03481777)	1500	Patients with acute stroke symptoms within 4 h of ictus	5 × 5 min inflations and deflations of cuff on one upper limb Cuff pressure: 200 mmHg or 35 mmHg above systolic pressure Times: once prehospital and twice daily for 7 days in hospital	RIC is safe but may not improve functional outcome at 90 days in patients with acute stroke
RESCUE BRAIN (NCT02189928)	188	Patients of AIS within 6 h of ictus	4 × 5 min inflation and deflation of cuff on one arm Cuff pressure: 110 mmHg above systolic pressure Times: Once prehospital	RIC, as an adjunct therapy of reperfusion, cannot limit brain infarction volume growth at 24 h after symptom onset.
SERIC-EVT (NCT04977869)	498	Patients with AIS underwent EVT	Ischemia for 5 min, reperfusion for 5 min, and repeat for 4 cycles Cuff pressure: 200 mmHg or 60 mmHg above systolic pressure Times: twice a day for 7 days after EVT	Not available for study; it is ongoing
REMOTE-CAT (NCT03375762)	572	Patients with suspected clinical stroke within 8 h of symptom onset	5 × 5 min inflations and deflations of cuff on an upper limb Cuff pressure: 200 mmHg or 0 mmHg above systolic pressure Times: within 48 h of symptom onset	RIC may increase the proportion of patients with good outcomes at 90 days
Hougaard et al. (2014) [49]	247/196	Patients with suspected acute stroke	4 × 5 min inflations or deflations of cuff on one upper limb Cuff pressure: 200 mmHg or 25 mmHg above systolic pressure Times: once prehospital	RIC is safe, feasible, and may reduce tissue infarction risk
NBO				
Li et al. (2022) [62]	43/43	Patients with anterior AIS undergoing EVT	100% O_2_, 10 L/min by face mask or routine low-flow oxygen by nasal cannula for 4 h, before EVT	NBO combined with EVT is safe and reduces the infarct volume in the early stage after ictus
Cheng et al. (2022) [65]	44/43	Patients with posterior stroke after EVT	50% O_2_, 15 L/min by Venturi mask or routine low-flow oxygen by nasal cannula, after EVT for 6 h	High-flow adjuvant NBO is safe but does not improve the mRS at 90 days
Li et al. (2021) [64]	125/102	Patients with anterior AIS received IVT with 4.5 h of ictus	70–80% O_2_, 10 L/min by high concentration oxygen mask or routine low-flow oxygen by nasal cannula for 4 h	NBO combined with IVT is safe and may improve functional outcomes at 90 days
Cheng et al. (2021) [63]	91	Patients with anterior stroke after EVT	50% O_2_, 15 L/min, by Venturi mask, after EVT for 6 h	High-flow adjuvant NBO is safe and may improve the functional outcomes at 90 days
Shi et al. (2017) [91]	18	Patients with acute ischemic stroke	10 L/min for 4 h after diagnosis of AIS	NBO therapy improves neurological functions in patients with AIS
Mazdeh et al. (2015) [92]	52	Patients with severe acute stroke	50% O_2_, by Venturi mask for first 12 h of admission	NBO could improve long-time outcomes of patients with stroke
SPGS				
Saver et al. (2019) [71]	50	Patients with anterior AIS, including arm weakness within 24 h of onset	4 h daily stimulation for 5 days	SPG stimulation improved brain blood flow, vessel diameter, and flow velocity and decreased hand motor weakness
ImpACT-24B (NCT00826059)	1078	Patient’s anterior circulation AIS after 8–24 h ictus	4 h daily stimulation for 5 days	SPGS is safe, and patients with CCI may benefit from SPGS
Bornstein et al. (2019) [93]	303	Patients with anterior AIS within 24 h of onset	4 h daily stimulation for 5 days	SPGS is safe, and patients with CCI may benefit from SPGS
ImpACT-1 (NCT03733236)	98	Patients with anterior AIS within 24 h from stroke onset	3–4 h per day for 5–7 days	SPGS is safe, feasible, tolerable, and effective to improve functional outcomes at 90 days
SBC				
Choi et al. (2010) [73]	18	Patients undergoing follow-up cerebral angiography after previous treatment of vascular malformations	4–7 °C or 12–17 °C saline infused into internal carotid artery at 33 mL/min for 10 min	1. JVB temperature drops 0.84 +/− 0.13 °C from baseline.2. Endovascular brain cooling is safe and feasible.
Chen et al. (2016) [74]	28	Patients with LVO within 8 h after onset and undergoing MT	4 °C, 50 mL saline at 10 mL/min infused pre- and post-reperfusion	1. Ischemic cerebral tissue was decreased by at least 2 °C.2. IA-CSI is safe and feasible to MT with LVO.
Peng et al. (2016) [75]	26	Patients with acute MCA occlusion	4 °C, 500 mL saline infused into internal carotid artery at 50 mL/min	Intra-arterial hypothermia is safe and effective in reducing infarct volume.
Wu et al. (2018) [76]	113	Patients with LVO-induced acute ischemic stroke and undergoing MT	IA-CSI with 350 mL saline at 4 °C for 15 min pre- and post-reperfusion	IA-CSI is effective in reducing infarct volume but cannot improve the functional outcomes at 90 days.

Abbreviations: RIC, remote ischemic conditioning; AIS, acute ischemic stroke; EVT, endovascular thrombectomy; NBO, normobaric hyperoxia; mRS, modified Rankin scores; IVT, intravenous thrombolysis; SPGS, sphenopalatine ganglion stimulation; SPG, sphenopalatine ganglion; CCI, confirmed cortical involvement; MCA, middle cerebral artery; LVO, large-vessel occlusion; MT, mechanical thrombectomy; IA-CSI, intra-arterial selective cooling infusion; JVB, jugular venous bulb.

## Figures and Tables

**Figure 1 biomolecules-14-01181-f001:**
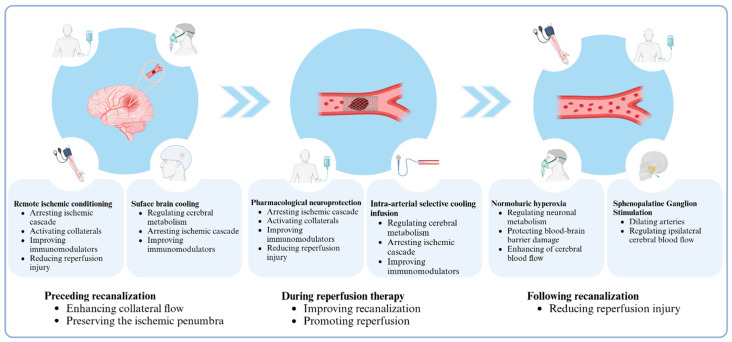
Multi-target and multi-phase adjuvant neuroprotection for acute ischemic stroke in the reperfusion era.

## Data Availability

Not applicable.

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
