# Peer review of "Multi-Target and Multi-Phase Adjunctive Cerebral Protection for Acute Ischemic Stroke in the Reperfusion Era"

_biomolecules, 2024, doi:10.3390/biom14091181_

Round 1

Reviewer 1 Report

Comments and Suggestions for Authors

In the commentary manuscript titled "Multi-target and Multiphase Adjunctive Cerebral Protection for Acute Ischemic Stroke in the Reperfusion Era," the authors examine why the high success rate of revascularization therapy does not consistently lead to equally impressive functional outcomes for patients with acute ischemic stroke. By analyzing the complex mechanisms underlying reperfusion injuries, the authors advocate for an adjunctive neuroprotective approach. This approach, designed to complement the primary recanalization strategy, targets multiple pathophysiological pathways and is applied at various time points to enhance patient outcomes.

This well-structured work serves as an updated, concise, and informative guide for researchers, facilitating access to essential review articles and clinical studies on this topic. The idea of revitalizing previously ineffective neuroprotective strategies as adjuncts to recanalization therapies for AIS is well-received. However, the concept of multi-target and multiphase approaches is debatable.

Firstly, the authors should clearly define the multiple phases mentioned in the text. It is recommended to visualize these phases on a timeline.

Regarding the concept of multiple targets, two arguments arise:

1. The multi-target approach suggests that combining several promising therapies should lead to better outcomes. However, this principle overlooks potential adverse interactions between therapies.

2. Advocating for the multi-target approach may lead to misunderstandings, justifying the use of unapproved, non-selective alternative treatments that often lack specific pharmacological targets. These treatments are frequently rationalized with claims of multi-target pharmacodynamics.

The authors should address these concerns in the discussion. Additionally, some fact-checks are needed. Lines 34 to 36 state, “Endovascular thrombectomy could achieve a favorable recanalization rate of up to 88% or higher, greatly improving patients’ functional outcomes [6-8].” The authors are encouraged to write a response letter in which they should clarify the data sources for the 88% figure from references 6 to 8. Similarly, lines 38 to 40 state, “with approximately 50-60% of patients experiencing poor functional outcomes or death within 90 days after reperfusion treatments [9,10].” The authors should clarify the sources for the 50-60% figure from references 9 and 10.

General suggestions include:

- Lines 45 to 46 cite references 13 and 14 to highlight the disappointing outcomes of cerebral protective treatments in clinical settings. Both references address hypothermic therapy. Considering the diversity of neuroprotectants, could the authors include any characteristic pharmacological neuroprotectants with failed clinical trials reported in high-impact journals?

- Lines 55 to 57 summarize the main targets of neuroprotective strategies based on references 15 and 16. While reference 15 should be highlighted for its focus on pharmacological brain cytoprotection, reference 16 primarily deals with the no-reflow phenomenon. The authors might consider omitting reference 16.

- Lines 57 to 58 reference 17, a review paper on stroke prevention among AF patients, is included as a historical summary of cytoprotective therapeutic trials. The authors should explain this choice.

- Lines 58 to 59 state that single-target neuroprotectants have limited efficacy in improving functional outcomes. In contrast, reference 18 endorses the promising potential of the neuroprotectant activated protein C (APC) for clinical trials.

Other specific suggestions include:

- Line 67: Replace "is" with "may be associated with" to avoid a definitive tone.

- Line 78: Reference 25 discusses mental practice as an option for post-stroke rehabilitation, which is more related to synaptogenesis and neural network remodeling/rewiring than neuroprotection.

- Correct the repeated typo "NHISS" to "NIHSS" in lines 103 and 115.

- Line 226: Check the typo “transnasal cooling ha?”

- Line 217: Clarify the meaning of “TH.”

- Lines 237 to 238: Explain how reference 67, a biomarker study on secondary prevention of ischemic stroke, is relevant to multi-target pharmacological drugs.

- In Table 1, clarify the meaning of “tape of patients.” Also, the column name “RIC protocol” should be revised as it includes other neuroprotective strategies. Verify the protocol for the RESCUE BRAIN study: the cuff pressure of 110 mmHg above systolic pressure. There should be some errors.

- Perform grammatical checks throughout the manuscript, particularly in lines 137, 182, 250, 251, and the main results colume of the last two rows in Table 1.

Comments on the Quality of English Language

The writing is perfect, except for some typos and grammar concerns.

Reviewer 2 Report

Comments and Suggestions for Authors

The Commentary "Multi-target and Multiphase Adjunctive Cerebral Protection for Acute Ischemic Stroke in the Reperfusion Era" discusses the advantages and disadvantages of different options for neuroprotective therapy of stroke after reperfusion. The review is pretty well structured, but lacks specificity. There are many inaccuracies in the review, and the text does not provide transcriptions of abbreviations, making it difficult to understand the material.  

1. Introduction

Acute ischemic stroke – it is necessary to specify that hereinafter the abbreviation (AIS)  is used 29

Cerebral protection has been 42

considered the most promising strategy, and thousands of neuroprotective methods have 43

been developed or discovered, hundreds of them being translated into clinical practice. – Links to relevant review publications should be added

2. Reasons for Mediocre Prognosis

NA-1 was once regarded as a promising neuroprotective drug to improve clinical outcomes for 60

patients with AIS [15]. – provide the drug's own name: nerinetide

The primary reason for the failed trials is the absence of comprehensive and effective 67

neuroprotection treatment strategies that target the multi-faceted pathophysiological 68

processes of AIS and the various phases of reperfusion therapy while integrating with 69

revascularization therapy [15,23,24]. – Can you be more specific about which variants of strategies - multi-faceted pathophysiological or phases of reperfusion therapy - do you mean?

3.1. Remote Ischemic Conditioning

injury and concurrently enhance cerebral flow through various pathways that encompass 85

humoral, neural, and immune mechanisms [27,28] – Give specific examples of the mechanisms discussed.

Recently, 92

several large-scale randomized clinical trials have been successfully conducted, focusing 93

on exploring the efficacy of RIC in AIS, and the results have been encouraging, 94

significantly enhancing the likelihood of achieving excellent neurological function [32,33] – insert the number and reference to studies from https://clinicaltrials.gov/. Do the same further down the text, where possible, in sections 3. Nonpharmacological Neuroprotective Methods and 4. Pharmacological Neuroprotective Methods

Add literature references in table 1 .

The article and table do not provide transcriptions, which makes it difficult to understand.

Is this what was meant?:

TH                              «However, clinical trials have rarely achieved the same level of success as TH in animal» unknown abbreviation

NHISS                        the National Institute of Health Stroke Scale https://www.ncbi.nlm.nih.gov/pmc/articles/PMC9406250/

RICAMIS                  The Remote Ischemic Conditioning for Acute Moderate Ischemic Stroke https://www.ncbi.nlm.nih.gov/pmc/articles/PMC9382441/

RESIST                      The Remote Ischemic Conditioning in Patients With Acute Stroke Trial  https://www.ncbi.nlm.nih.gov/pmc/articles/PMC10548297/

SERIC-EVT               The safety and efficacy of RIC combined with endovascular thrombectomy (SERIC-EVT) https://pubmed.ncbi.nlm.nih.gov/35971654/

SERIC-IVT                … with intravenous thrombolysis https://pubmed.ncbi.nlm.nih.gov/35619218/

RCT                            Randomized control trial https://pubmed.ncbi.nlm.nih.gov/?term=RCT

PROOF                      Penumbral Rescue by normobaric O = O administration in patients with ischemic stroke and target mismatch profile https://www.ncbi.nlm.nih.gov/pmc/articles/PMC10759237/

ImpACT-24B             IMPlant Augmenting Cerebral blood flow Trial-24B https://doi.org/10.1016/S0140-6736(19)31192-4

RESCUE BRAIN      The Remote Ischemic Conditioning in Acute Brain Infarction https://www.ncbi.nlm.nih.gov/pmc/articles/PMC7105950/

REMOTE-CAT          unknown abbreviation. Found link on the web https://pubmed.ncbi.nlm.nih.gov/33101178/

The latter two are not mentioned in the text and there are no references in the table.

In the Commentary, the authors talk about the concept of "multi-target and multiphase adjunctive neuroprotection", but nowhere do they offer specific schemes of such therapy. In my opinion, the Conclusion should contain possible variants of such treatment schemes, which, according to the authors' opinion, will be the most effective.

Round 2

Reviewer 1 Report

Comments and Suggestions for Authors

I appreciate the significant efforts the authors have put into revising the manuscript. Most of my concerns seem to have been addressed. However, I would like to add a few more comments:

1. Figure 1 is well prepared, but a high-resolution version is necessary. When zoomed in, either in the author’s reply or in the revised manuscript, the text in the figure remains blurry.

The figure design displays several promising therapeutic options in each phase in parallel. This presentation may give readers the impression that these multiple options are intended to be prescribed simultaneously. Therefore, the authors are encouraged to clarify in the text that current clinical trials assess only one therapeutic option at a time. The potential benefits of combining these options have yet to be investigated in humans (unless there is any trial addressing this?).

Another concern arises from the middle panel of Figure 1. The authors are invited to conclude, based on clinical trials, whether these neuroprotective options are feasible during the exact moment of reperfusion therapy. Are any of them, such as IA-CSI, administered simultaneously with thrombolysis or EVT?

2. Lines 262 to 263: The sentence reads, "However, clinical trials have rarely achieved the same level of success as hemorrhagic transformation in animals." I do not understand how a hemorrhagic transformation complication in animals would be a success that should be replicated in patients. The authors are invited to reevaluate this sentence based on its context.

Reviewer 2 Report

Comments and Suggestions for Authors

The review " Commentary «Multi-target and Multiphase Adjunctive Cerebral Protection  for Acute Ischemic Stroke in the Reperfusion Era»" discusses the advantages and disadvantages of different options for neuroprotective therapy of stroke after reperfusion. There are no complaints about the structure of the narrative. Some deficiencies were noted in the text that need to be corrected:

1. The previous reviewer comment: "The primary reason for the failed trials is the absence of comprehensive and effective 67

neuroprotection treatment strategies that target the multi-faceted pathophysiological 68

processes of AIS and the various phases of reperfusion therapy while integrating with 69

revascularization therapy [15,23,24].  –   Can you be more specific about which strategy options - multidimensional pathophysiologic or step-reperfusion therapy - you mean?" 

– You put the path.processes of stroke in before this sentence (that's good), but you didn't elaborate here what strategy options were meant.

2. "However, the ESCAPE-NA1 () study revealed that while the administration of ..." (line 64)

– insert after ESCAPE-NA1 a link to the relevant page of the study from https://clinicaltrials.gov/.

3. "with EVT () did not enhance clinical outcomes when compared to .. " (line 71)

– Decipher the EVT acronym here

4. "Their funding revealed that this... "  (line 119)

– Do you mean finding?

5. "remote ischemic conditioning (RIC).[42] " (line 123.)

– The acronym is already transcribed above

6. "among different NHISS score. Furthermore, TRICS-9 ."  (line 155)

– mistype??? Supposed to be "NIHSS" as above

7. "The sphenopalatine ganglion(), a crucial structure for intracranial blood vessel " (line 195)

– The SPG acronym should be introduced here, not below

8. "stimulating the sphenopalatine ganglion  (SPG)" (line 200)

–  See the previous notice

9. "Figure 1" (line 328) 

– Very tiny text in the figure itself, unreadable.
